# Determinants of short birth interval among ever married reproductive age women: A community based unmatched case control study at Dessie city administration, Northern Ethiopia

**Habtamu Shimels Hailemeskel**[1]*, **Tesfaye Assebe**[2], **Tadesse Alemayehu**[3], **Demeke Mesfin Belay**[1], **Fentaw Teshome**[4], **Alemwork Baye**[5], **Wubet Alebachew Bayih**[1]

1 Department of Nursing, College of Health Sciences, Debre Tabor University, Debre Tabor, Ethiopia,
2 School of Nursing and Midwifery, College of Health and Medical Sciences, Haramaya University, Harar, Ethiopia, 3 School of Public Health, College of Health and Medical Sciences, Haramaya University, Harar, Ethiopia, 4 Department of Health Service Management, Debre Tabor University, Debre Tabor, Ethiopia, 5 Department of Environmental Health, Wollo University, Dessie, Ethiopia

* habtamushimels21@gmail.com

## Abstract

### Background

Short birth interval is a universal public health problem resulting in adverse fetal, neonatal, child and maternal outcomes. In Ethiopia, more than 50% of the overall inter birth spacing is short. However, prior scientific evidence on its determinants is limited and even then findings are inconsistent.

### Methods

A community -based unmatched case-control study was employed on 218 cases and 436 controls. Cases were ever married reproductive age women whose last delivery has been in the past five years with birth interval of less than 3 years between the latest two successive live births whereas those women with birth interval of 3–5 years were taken as controls. A multistage sampling technique was employed on 30% of the kebeles in Dessie city adminis-tration. A pre-tested interviewer based questionnaire was used to collect data by 16 trained diploma nurses and 8 health extension workers supervised by 4 BSc nurses. The collected data were cleaned, coded and double entered into Epi-data version 4.2 and exported to SPSS version 22. Binary logistic regression model was considered and those variables with P<0.25 in the bivariable analysis were entered in to final model after which statistical signifi-cance was declared at P< 0.05 using adjusted odds ratio at 95% CI.

### Result

In this study, contraceptive use (AOR = 11.2, 95% CI: 5.95–21.15), optimal breast feeding for at least 2 years (AOR = 0.098, 95% CI:0.047–0.208), age at first birth <25 years (AOR =

**Data Availability Statement:** All relevant data are within the manuscript and its Supporting Information files.

**Funding:** The author(s) received no specific funding for this work.

**Competing interests:** The authors have declared that no competing interests exist.

**Abbreviations:** AOR, Adjusted odds ratio; EDHS, Ethiopian Demographics and Health Surveillance; FMOH, Federal Ministry of health; HEWs, Health Extension workers; LBW, Low birth weight; PROM, Premature rupture of membrane; WHO, World Health Organizations.

0.36, 95% CI: 0.282–0.761), having male preceding child (AOR = 0.46, 95% CI: 0.166–0.793) and knowing the duration of optimum birth interval correctly (AOR = 0.45, 95% CI: 0.245–0.811) were significant determinants of short birth interval.

## Conclusion

Contraceptive use, duration of breast feeding, age at first birth, preceding child sex and correct understanding of the duration of birth interval were significant determinants of short birth interval. Fortunately, all these significant factors are likely modifiable. Thus, the existing efforts of optimizing birth interval should be enhanced through proper designation and implementation of different strategies on safe breastfeeding practice, modern contraceptive use and maternal awareness about the health merits of optimum birth interval.

## Background

Birthinterval refers to the time gap between two consecutive live births [1]. In 2005, World Health Organization consultation meeting on pregnancy intervals recommended a minimum inter pregnancy interval of at least 24 months to reduce the risk of adverse maternal, perinatal, and infant outcomes [2]. Moreover, the Ethiopian national family planning guideline recommends spacing childbirth at intervals of three to five years to reduce adverse fetomaternal and neonatal complications [3].

Short birth interval is a universal public health problem having association with adverse maternal, fetal, neonatal and child outcomes such as low birth weight and perinatal death [4, 5], preterm delivery, small for gestational age [6], admission to neonatal intensive care unit [7], stillbirth, abortion, neonatal mortality [8], infant and under-5 mortality [8, 9], infant/child malnutrition including underweight, wasting, stunting [8, 10], neurodevelopmental and intellectual delay, autism, cerebral palsy [11], gestational diabetes [8, 12], precipitous labor [7], anemia [8, 13], uterine rupture, premature rupture of membrane, preeclampsia and chronic hypertension[8, 14, 15]. Most of these studies [4–7, 9–15] don't show causal association between short birth interval and the aforementioned pregnancy outcomes. Furthermore, the reported associations might have been largely attributed to confounding effects by genetically heritable familial factors [16, 17]. On the contrary, a systematic review of the available literature about the effects of birth spacing on maternal, perinatal, infant and child health witnessed the presence of causal mechanisms of association between short interbirth interval and its predictors [8].

Ethiopia had high population size as it was projected to reach more than 100 million and 4.0 total fertility rates in 2015. The country had also higher estimated pregnancy-related mortality ratio (PRM) of 412 deaths per 100,000 live births. Moreover, 1 in every 35 children dies within the first month; 1 in every 21 children dies before celebrating the first birthday; and 1 of every 15 children dies before reaching the fifth birthday (16). Therefore, the Ethiopian Federal Ministry of Health (FOMH) recommends spacing of childbirth at intervals of three to five years to reduce maternal, perinatal and infant mortality by optimizing the fertility rate in the country. However, in Ethiopia, more than 50% of the pregnancies occur within 3 years of their prior birth [18] which is shorter than the national recommendation of at least 3 years. Though initiatives like comprehensive implementation of family planning has been undertaken by the federal ministry of health at all levels of the health care system [3], the problem is of still

greatest concern. This is so because birth intervals vary from society to society and within society itself within a country population [19, 20].

Since short birth interval is a potentially modifiable problem, a better knowledge and understanding of its determinants is imperative and essential to improve maternal health by designing and applying specifically targeted interventions thereby decreasing catastrophic pregnancy outcomes [9, 16].However, evidence on the determinants of short birth interval in the study area is limited and even the nationally available data are inconsistent. Therefore, this study was aimed at identifying factors that have significant odds of association with short inter-birth interval among a community-based sample of Ethiopian women in Dessie city administration, 2019.

## Methods

### Study setting and period

Dessie city administration is located in northern part of Ethiopia at a distance of 401 km from Addis Ababa, capital of the country. It has an altitude of 2470 meters above sea level, situated between Tosa and Azewa mountains at11˚ 05´ North latitude and 39˚ 40´East longitude. The city administration has 5 sub cities. Besides, for administrative sake, the city is categorized into 18 urban and 8 rural kebeles (the lowest administrative levels in the study area). Based on the 2014 Ethiopian population projection, Dessie district had a total population of 212,436 of whom 83.6% (177,688) lived in urban areas [19]. The study was held from 5/1/2019-12/5/2019.

### Study design and participants' characteristics

A community based unmatched case-control study was conducted on a sample of eligible cases and controls. All the ever married reproductive age women who had at least two consecutive live births and whose last delivery within the past five years before the survey were eligible for the study.

The eligible women who had history of less than 3 years birth interval between their two successive live births were considered as cases. Besides, controls were considered to be those eligible women with birth interval of 3–5 years (including 3 and 5) between their two successive live births.

### Sample size determination and sampling procedure

Taking several exposure variables into account, we calculated the respective sample size just by considering the assumption of case to control ratio of 1: 2; CI: 95%; Power: 80%; minimum detectable AOR = 2; design effect of 1.5 and 5% non-respondent rate. Among the given factors, we selected 'contraceptive use' because it yielded the maximum sample size as given in the following table (Table 1). Therefore, the final sample size was 678 (226 cases and 452 controls).

Then, multi stage sampling technique was employed to select the cases and controls. At first, 30% of the overall 'kebeles' (three rural and five urban kebelles), were selected by simple random sampling technique. For those rural kebeles, the authors first checked family folder from health extension workers. We reviewed the family folder of permanently residing women in each kebele that fulfilled the inclusion criteria (less than 3 years birth intervalfor cases and 3–5 years' birth interval (including 3 and 5 years for controls)) by registering the birth date of the last two successive children in a family with their corresponding household identification number. However, for urban 'kebeles', house to house visit (census) was conducted to identify permanently residing women that fulfilled the inclusion criteria (cases and controls) by registering the birth date of the last two successive children in a family with their corresponding

**Table 1. Sample size determination involving different factors in the literature and the respective assumptions using open EPI INFO version 7 software.**

| Factors | Assumption | Total sample size | References |
|---|---|---|---|
| Contraceptive user | P of exposure in controls = 66.7% | 678 | (Hailu and Gulte, 2016) |
| Residence/urban | P of exposure in controls = 52.1% | 540 | (Yohannes et al., 2011) |
| Husbands' occupation /Employee | P of exposure in controls = 51.7% | 537 | (Yohannes et al., 2011) |
| Mothers' education /Has formal education | P of exposure in controls = 48.3% | 524 | (Hailu and Gulte, 2016) |
| Parity /> = 5 children | P of exposure in controls = 49.2% | 524 | (Begna Z. et al., 2013) |
| Sex of the index child /male | P of exposure in controls = 64.2% | 638 | (Begna Z. et al., 2013) |
| Age of the mother/ 25–29 | P of exposure in controls = 24.9% | 576 | (Begna Z. et al., 2013) |
| Status of index child /Alive | P of exposure in controls = 41.3% | 509 | (Tsegaye Dereje et al., 2017) |
| Wealth index/ Richest | P of exposure in controls = 25.2% | 509 | (Hailu and Gulte, 2016) |

household identification number. Using the respective household identification number, a sampling frame of the households containing cases and controls was prepared for each kebele. Then, proportional allocation of sample size was employed to determine the study participants from each kebele. Finally, cases and controls were selected by simple random sampling technique from the existing sampling frame. Whenever more than one eligible woman was found in same selected household, only one woman was chosen by lottery method. Thus, a sample of 678 women (226 cases and 452 controls) was recruited from the sampling frame for the study (Fig 1).

## Measurement and data collection procedure

Using interviewer based questionnaire, eight heath extension workers and sixteen diploma nurses underwent the data collection process including the weekend. During data collection, out of 654 eligible women (218 cases and 436 controls), 24 eligible women (8 cases and 16 controls) weren't accessed even after 2 different return visits. Therefore, these 24 absentees were replaced by other 24 randomly selected eligible mothers. The replaced mothers weren't systematically different from the original mothers because the replaced mothers were randomly selected from the already prepared sampling frame of eligible mothers (i.e. volunteers weren't included).Then, all the selected cases and controls were approached to be interviewed about factors related to their socio-demography, obstetrics, breastfeeding practice and modern contraception. Besides, the respondents were asked about their knowledge and attitude of birth interval. To determine children's birth dates, birth certificate or immunization cards were used. For those who were not immunized, health extension workers or mother's memory was consulted.

## Data quality control

A structured English version interviewer based questionnaire (S1 Questionnaire) was first adapted from different literatures [1, 16, 20–22] and then translated to Amharic version (local language) for data collection purpose. The questionnaire was pretested just two weeks prior to the actual data collection using 33 eligible women (5% of the sample size) at the study area based on which some modifications were made to the originally prepared tool. Data collectors were closely monitored and guided by four BSC nurse supervisors. There was no missing information for any of the covariates in this study. This was because incomplete questionnaires were returned to the data collectors for completion by referring to the respective household identification number on a daily basis of checking all the questionnaires.

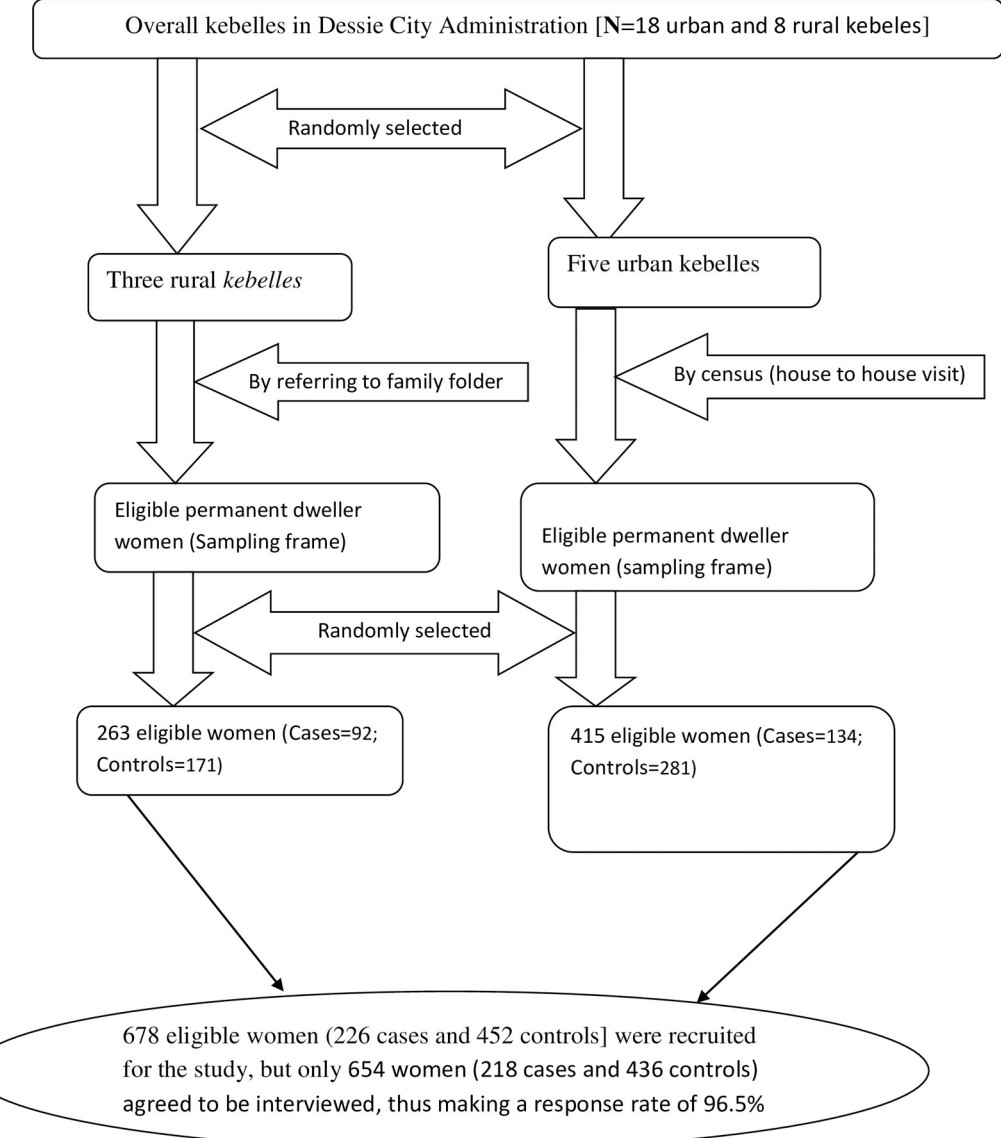

**Fig 1. A flow diagram of sampling procedure.**

## Data processing and analysis

Data were coded and double entered into Epi-Data software version 4.2 and then exported to SPSS version 22 for further processing and analysis. Descriptive statistics of different variables was done by cross tabulation. Binary logistic regression model using bivariable [crude odds ratio, [COR] and multivariable analyses [adjusted odds ratio, AOR] with 95% Confidence interval [CI] was employed. During bivariable analysis, variables whose $p<0.25$ were reserved for inclusion into the multivariable analysis in the final model after which statistical significance was declared at $P< 0.05$ using adjusted odds ratio. Both Hosmer-Lemeshow's test (p = 0.753) and Omnibus Tests (p = .000) were used to check model fitness. Multi-collinearity was checked to see the linear correlation among the independent variables by using variance inflation factor and standard error. It was tried to minimize bias from intra-cluster correlation effect (dependencies) by considering only one of the eligible women in a selected household.

Besides, standard error was used during multivariate regressions and there was no any factor whose standard error greater than two indicating no dependency between mothers regarding the considered factors.

**Estimation of household wealth index.** Wealth index of the studied households were given scores based on the number and kinds of consumer goods they own including chairs, tables, chicken, transport (vehicles) and household characteristics like source of drinking water, toilet facilities, wall, roof and flooring materials. Among the nine characteristics, eight of them were extracted.

SPSS version 22 software was used to perform principal component analysis (PCA). Finally, wealth status was categorized into five groups and ranked from poorest to wealthiest quintile. Kaiser-Meyer-Olkin Measure of Sampling Adequacy was 0.751 and Bartlett's Test of Sphericity was significant.

*Ethical approval and consent to participate.* Ethical approval with ethics approval number of HU-CHMS-001 was obtained from Haramaya University, College of Health and Medical Sciences, Institutional Health Research Ethics Review Committee (IHRERC). An informed and voluntarily signed written consent (thumb print for those unable to write) was obtained from all the eligible mothers. Parental consent wasn't required because all the respondent mothers were above 16 years of old.

## Results

### Socio-demographic characteristics

From the overall sample of 678 mothers, 654 women (218 cases and 436 controls) agreed to be interviewed, thus making a response rate of 96.5%. Median age of the respondents at last delivery was 32 years. Twenty four (11%) of the cases and 82 (18.8%) of the controls were married at their age of 18 or less years. One hundred thirty five (61.9%) of the cases and 287 (65.8%) of the controls were within the age of25–34 years. Regarding their residence, 130 (59.6%) of the cases and 273 (62.6%) of the controls were urban residents. Nearly one fourth of the cases 58 (26.6%) and controls 110 (25.2.0%) had college and above level of education. One hundred twenty five (57.3%) of the cases and 232 (53.2%) of the controls were house wives. Moreover, 37(17.0%) of the cases and 107(24.5%) of the controls had the richest wealth index (Table 2).

### Knowledge and attitude on birth interval

One hundred sixty five (75.7%) of the cases and 352 (80.7%) of the controls had ever heard about optimal birth interval. One hundred thirty four (61.5%) of the cases and 291 (66.7%) of the controls agreed that a minimum of 3 years birth spacing is essential between two successive births. Regarding husbands' perception of birth spacing, 120 (55%) of the cases and 246 (56.4%) of the controls had encouraging perception to birth spacing. One hundred forty four (66.1%) of the cases and 298(68.3%) of the controls had nobody to influence them to give birth with short interval. Two hundred and four (93.6%) of the cases and 404(92.7%) of the controls perceived that short birth interval have disadvantages on both maternal and child health. Regarding respondents' knowledge of the optimum birth interval, 130(78.8%) of the cases and 280(79.5%) of the controls knew the appropriate cut point correctly. The source of information for majority of the cases 112(67.9%) and controls 289(82.1%) were health workers (Table 3).

### Obstetrics related factors

The mean maternal age at first birth was 23(±3.47) years. The median length of time from marriage to first birth was 24 months. Equal proportion (12.4%) of the cases and controls had

**Table 2. Socio-demographic characteristics on short birth interval among ever married mothers (case = 218, control = 436) in Dessie city administration, Dessie, Ethiopia 2019.**

| Factors | Category | Case (%) | Control(%) | P value |
|---|---|---|---|---|
| Rsidence | Urban | 130(59.7%) | 273(62.65) | 0.460 |
| | Rural | 88(40.3%) | 163(37.4%) | |
| Marital status | Married | 186(85.3%) | 364(83.5%) | 0.759 |
| | Divorced | 21(9.6%) | 44(10.1%) | |
| | Widowed | 11(5.1%) | 28(6.4%) | |
| Religion | Orthodox | 92(42.2%) | 173(39.7%) | 0.287 |
| | Muslim | 124(56.9%) | 249(57.1%) | |
| | Protestant | 2(0.9%) | 14(3.2%) | |
| Ethinicity | Amhara | 200(91.7%) | 399(91.5%) | 0.926 |
| | Tgrai | 7(3.2%) | 11(2.5%) | |
| | Oromo | 6(2.7%) | 14(3.2%) | |
| | Others[1] | 5(2.3%) | 12(2.8%) | |
| Mother's education | No formal education | 45(20.6%) | 70(16.1%) | 0.546 |
| | read and write | 42(19.3%) | 86(19.7%) | |
| | Elementary | 34(15.6%) | 81(18.6%) | |
| | Secondary | 39(17.9%) | 89(20.4%) | |
| | Collage and above | 58(26.6%) | 110(25.2%) | |
| Husband education | No formal education | 50(22.9%) | 69(15.8%) | 0.104 |
| | read and write | 32(14.7%) | 69(15.8%) | |
| | Elementary | 13(5.9%) | 42(9.6%) | |
| | Secondary | 41(18.8%) | 72(16.5%) | |
| | College and above | 82(37.6%) | 184(42.2%) | |
| Mothers' occupation | employee(GO/NGO) | 43(19.7%) | 91(20.9%) | 0.730 |
| | house wife | 125(57.3%) | 232(53.2%) | |
| | Merchant | 28(12.8%) | 53(12.2%) | |
| | Student | 9(4.1%) | 29(6.7%) | |
| | Farmer | 10(4.6%) | 19(4.4%) | |
| | daily workers | 3(1.4%) | 11(2.5%) | |
| | Others[2] | 0(0%) | 1(0.2%) | |
| Husband occupation | employee(GO/NGO) | 84(38.5%) | 164(37.6%) | 0.086 |
| | Merchant | 66(30.3%) | 129(29.6%) | |
| | Student | 0(0%) | 2(0.5%) | |
| | Farmer | 63(28.9%) | 107(24.5%) | |
| | daily workers | 4(1.8%) | 23(5.3%) | |
| | Others[3] | 1(0.5%) | 11(2.5%) | |
| Number of wives wealth index | One | 216(99.1%) | 434(99.5%) | 0.478 |
| | More than one | 2(0.9%) | 2(0.5%) | |
| | Poorest | 57(26.1%) | 84(19.3%) | 0.096 |
| | Second | 35(16.1%) | 80(18.3%) | |
| | Middle | 47(26.6%) | 83(19.0%) | |
| | Fourth | 42(19.3%) | 82(18.8%) | |
| | Richest | 37(17.0%) | 107(24.5%) | |

[1]Afar, Gurage

[2] House servant,

[3]Religious leader

**Table 3. Knowledge and attitude of birth interval among ever married reproductive age mothers (case = 218, control = 436) in Dessie city administration, Dessie, Ethiopia 2019.**

| Factors | Category | Case (%) | Control (%) | P value |
|---|---|---|---|---|
| Heard about optimal birth interval | Yes | 165(75.7%) | 352(80.7%) | 0.336 |
| | No | 53(24.3%) | 84(19.3%) | |
| Optimum number of years between two successive births | Below three years | 19(11.5) | 46(13.1%) | 0.701 |
| | Three to five years | 130(78.8%) | 280(79.5%) | |
| | Above five years | 13(7.8%) | 23(6.5%) | |
| | I am not sure | 3(1.8%) | 3(0.8%) | 0.562 |
| A minimum of 3 years of birth interval is essential between two successive births | Strongly agree | 81(37.2%) | 139(31.9%) | |
| | Agree | 134(61.5%) | 291(66.7%) | |
| | no idea | 2(0.9%) | 3(0.7%) | |
| | Disagree | 1(0.5%) | 3(0.7%) | |
| Husband's perception regarding birth spacing | Disagree strongly | 28(12.8%) | 27(6.2%) | 0.001 |
| | don't mind | 57(26.1%) | 152(34.9%) | |
| | Encouraging | 120(55.04%) | 246(56.4%) | |
| | Unknown | 13(5.96%) | 11(2.5%) | |
| External influences to give birth in short interval | My family | 37(16.97%) | 61(13.99%) | 0.258 |
| | Mother in law | 21(9.63%) | 60(13.76%) | |
| | Father in law | 7(3.2%) | 12(2.75%) | |
| | Societies norm | 9(4.1%) | 5(1.1%) | |
| | None | 144(66.1%) | 298(68.4%) | |
| Perceived advantages of optimum birth spacing | Yes | 205(94.04%) | 406(93.1%) | 0.655 |
| | No | 13(5.96%) | 30(6.9%) | |
| Perceived disadvantages of short birth interval | Yes | 204(93.6%) | 404(92.7%) | 0.665 |
| | No | 14(6.4%) | 32(7.3%) | |

bad fetal outcome at first delivery. Among the respondents, 5% of the cases and 2.5% of the controls experienced neonatal mortality. Besides, 3.7% of the cases and 1.8% of the controls had experienced stillbirth in their life time. Twenty five (5.7%) of the cases and 9(4.1%) of the controls had high birth order of their preceding child. From the overall respondents, 38 (17.4%) of the cases and 34 (7.8%) of the controls reported that their previous pregnancy was unplanned.

Forty six (21.1%) of the cases and 49(11.2%) of the controls had not ANC follow up for their previous pregnancy. Twenty five (11.5%) of the cases and 39(8.9%) of the controls had home delivery of their previous and last children. Majority of the cases 197(90.4%) and controls 397(89.9%) had spontaneous vaginal delivery of their previous child. Twenty six (11.9%) of the cases and 61(13.9%) of the controls ever had history of postpartum complications during their previous to last deliveries. From these complications, bleeding was reported among 6 (23.1%) of the cases and 30(49.2%) of the controls. The median duration of resuming postpartum sexual activity was 45 days. From the total respondents, 16 (7.3%) of the cases and 44 (10.1%) of the controls ever had chronic diseases like hypertension and diabetic mellitus before their last childbirth. The median ages of last and preceding child were 17and 60 months respectively (Table 4).

## Breastfeeding and modern contraception related factors

Most of the cases 198(90.9%) breast fed their children for less than 24 months whereas 178 (40.8%) of the controls breastfed for at least 24 months. Moreover, more than half of the cases

**Table 4. Obstetrics related factors of short birth interval among ever married reproductive age mothers (case = 218, control = 436) in Dessie city administration, Dessie, Ethiopia 2019.**

| Factors | Category | Case (%) | Control (%) | P value |
|---|---|---|---|---|
| Fetal outcome of first delivery | Live birth | 191(87.6%) | 382(87.62%) | 0.352 |
| | still birth | 11(5.04%) | 13(2.98%) | |
| | Abortion | 3(1.4%) | 13(2.98%) | |
| | Neonatal mortality | 13(5.96%) | 28(6.42%) | |
| Prior history of infertility | Yes | 4(1.83%) | 3(0.69%) | 0.279 |
| | No | 214(98.17%) | 433(99.31%) | |
| Ever given birth to any child who died | Yes | 31(14.2%) | 58(13.3%) | 0.723 |
| | No | 187(85.8%) | 378(86.7%) | |
| Male to female ratio of living children | More than one | 71(32.6%) | 160(36.7%) | 0.355 |
| | One | 63(28.89%) | 135(30.96%) | |
| | Less than one | 49(22.48%) | 74(16.97%) | |
| | Males only | 15(6.9%) | 36(8.26%) | |
| | Females only | 20(9.17%) | 31(7.11%) | |
| Previous to last pregnancy is planned | Yes | 180(82.6%) | 402(92.2%) | 0.001 |
| | No | 38(17.4%) | 34(7.8%) | |
| Practice postpartum abstinence before the last child | Yes | 161(73.85%) | 359(82.3%) | 0.011 |
| | No | 57(26.15%) | 77(17.7%) | |
| Mode of delivery of previous to last birth | Vaginal delivery | 197(90.4%) | 392(89.9%) | 0.981 |
| | Cesarean section | 14(6.4%) | 29(6.7%) | |
| | Instrumental delivery | 7(3.2%) | 15(3.4%) | |
| ANC follow up in preceding pregnancy | Yes | 172(78.9%) | 387(88.8%) | 0.009 |
| | No | 46(21.1%) | 49(11.2%) | |
| Place of delivery of previous to last birth | Home | 25(11.5%) | 39(8.9%) | 0.308 |
| | Health institution | 193(88.5%) | 397(91.1%) | |
| Pattern of menstruation in previous to last deliveries | Regular | 185(84.9%) | 362(83.02%) | 0.550 |
| | Irregular | 33(15.1%) | 74(16.97%) | |
| Ever had chronic diseases (HTN, DM, others) before the last child | Yes | 16(7.3%) | 44(10.1%) | 0.255 |
| | No | 202(92.7%) | 392(89.9%) | |
| Ever had history of postpartum complications in previous to last deliveries | Yes | 26(11.9%) | 61(13.99%) | 0.464 |
| | No | 192(88.1%) | 375(86.01%) | |
| Last child sex | Male | 121(55.5%) | 238(54.6%) | 0.824 |
| | Female | 97(44.5%) | 198(45.4%) | |
| Is last child alive | Yes | 217(99.5%) | 434(99.5%) | 0.741 |
| | No | 1(0.5%) | 2(0.5%) | |
| previous to last child sex | Male | 72(33%) | 235(53.9%) | 0.001 |
| | Female | 116(53.2%) | 201(46.1%) | |
| Is previous to last child alive | Yes | 215(98.6%) | 434(99.5%) | 0.254 |
| | No | 3(1.4%) | 2(0.5%) | |
| Parity | <5 | 180 (82.5%) | 370(84.8%) | 0.450 |
| | > = 5 | 38(17.5%) | 66(15.2%) | |

80 (52.6%) and three fourth of the controls 295(73.8%) practiced exclusive breastfeeding to their preceding child. Ninety eight (44.9%) of the cases and 411 (94.3%) of the controls have utilized modern contraceptive methods after delivering their preceding child. Nearly all of the cases 213 (97.7%) and 434(99.5%) of the controls knew at least one type of modern contraceptive. One hundred eighty three (83.9%) of the cases and 428(98.2%) of the controls agreed that

family planning method is necessary for birth spacing. Regarding decision making about family planning in the house hold, ninety seven (44.5%) of the cases and 227(52.1%) of the controls decided based on couple agreement (Table 5).

Concerning the practice of modern contraceptive methods, forty three (43.9%) of the cases and 183(44.5%) of the controls utilized injectable type after delivering their preceding child (Fig 2).

## Determinants of short birth interval

From the total fourteen variables that were entered to the multivariable logistic regression analysis, only five of them namely contraceptive use (AOR = 11.2, 95% CI: 5.95–21.15), optimal breast feeding for at least 2 years (AOR = 0.098, 95% CI:0.047–0.208), age at first birth<25 years (AOR = 0.36, 95% CI: 0.282–0.761), having male preceding child (AOR = 0.46, 95% CI: 0.166–0.793) and knowing the duration of optimum birth interval correctly (AOR = 0.45, 95% CI: 0.245–0.811) had significant odds of association with short birth interval. We used backward stepwise method to identify variables which had the largest contribution to the regression model. The result in forward or a stepwise variable selection method was similar on significance of the variables, but little change in adjusted odds ratio, p value and confidence interval were observed.

The odds of short birth interval among mothers who breastfed their prior child for at least 24 months were 90.2% lower (AOR = 0.098, 95% CI: 0.047–0.208) as compared to those having less than 12 months of breastfeeding duration. The odds of short birth interval among mothers having male preceding child was 54.0% lower than those whose child was female (AOR = 0.46, 95% CI: 0.166–0.793). Besides, the odds of short birth interval among those who didn't use modern contraceptives was11.2 times higher as compared to the users (AOR = 11.22, 95% CI: 5.95–21.15). Concerning maternal knowledge about the duration of birth interval, those mothers who knew the duration correctly had 55% lower odds of association with short birth

**Table 5. Breast feeding duration and contraceptive use among ever married reproductive age mothers in Dessie city administration, Dessie, Ethiopia 2019.**

| Factors | Category | Case (%) | Control (%) | P value |
|---|---|---|---|---|
| Did you breast feed previous to last child | Yes | 152(69.7%) | 400(91.7%) | 0.001 |
| | No | 66(30.3%) | 36(8.3%) | |
| Did you exclusively breastfeed previous to last child | Yes | 80(52.6%) | 295(73.8%) | 0.001 |
| | No | 72(47.4%) | 105(26.2) | |
| Breast feeding duration | 0–11 | 134(61.5%) | 61(13.99%) | 0.001 |
| | 12–23 | 64(29.4%) | 197(45.18%) | |
| | > = 24 | 20(9.2%) | 178(40.83%) | |
| Using any of the modern methods before the conception of your last child | Yes | 98(44.95%) | 411(94.3%) | 0.001 |
| | No | 120(55.05%) | 25(5.7%) | |
| Decision maker about Family planning | Self | 104(47.7%) | 190(43.58) | |
| | Both husband and wife | 97(44.5%) | 227(52.06%) | 0.261 |
| | Husband only | 3(1.4%) | 13(2.98%) | |
| | No one | 14(6.4%) | 6(1.38%) | |
| Perception of family planning method | Agree | 183(83.9%) | 428(98.2%) | 0.001 |
| | Disagree | 34(15.6%) | 4(0.9%) | |
| | Neutral | 1(0.5%) | 4(0.9%) | |
| Distance from health institution | Less than 30 minutes | 93(42.7%) | 197(45.2%) | 0.799 |
| | 30-1hrs | 123(56.4%) | 236(54.1%) | |
| | Greater than 1 hr | 2(0.9%) | 3(0.7%) | |

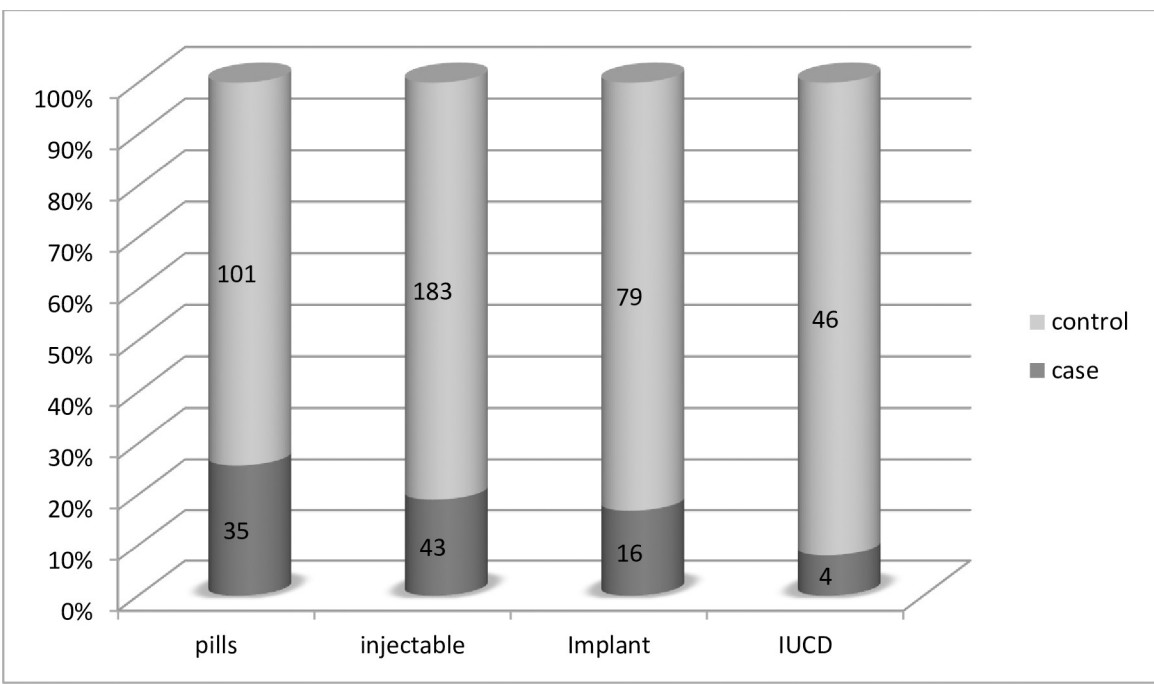

**Fig 2. Type of modern contraceptives utilized among ever married reproductive age mothers in Dessie city administration, Dessie, Ethiopia, 2019.**

interval (AOR = 0.45, 95% CI: 0.245–0.811) as compared to those who didn't know the duration correctly. Lastly, it was found that mothers who gave their first birth at the age of less than 28 years had 64% lower odds of association with short birth interval when compared to their counterparts (AOR = 0.46, 95% CI: 0.282–0.761) (Table 6).

Despite no statistical significance in the adjusted analysis, the crude odds of short birth interval was lower among mothers who had planned preceding pregnancy (COR = 0.401, 95% CI: 0.244–0.657) than those whose pregnancy wasn't planned. Besides, mothers who abstained in the post partum period had lower crude odds of short birth interval (COR = 0.61, 95% CI: 0.410–0.8941) than those who didn't abstain. Similarly, mothers who had ANC follow up [COR = 0.547, 95% CI: 0.348–0.858), mothers whose age at first marriage >25 years (COR = 5.256, 95% CI: 2.68–10.286), husband's encouraging perception of birth spacing (COR = 0.470, 95% CI: 0.266–0.833) and those mothers having the poorest wealth index (COR = 1.962, 95% CI: 1.187–3.245) were crudely associated with short birth interval (Table 6).

## Discussion

This study was employed to investigate the determinants of short birth interval among ever married reproductive age mothers at Dessie city administration. Thus, from the adjusted analysis, it was found that contraceptive use, breast feeding duration, age at first birth, preceding child sex and knowing the appropriate duration of optimum birth interval correctly were significant determinants of short birth interval.

In this study, not using modern contraceptive method before getting pregnant of the last child was positively associated with short birth interval as compared to the users. This finding is similar to studies in Kassala, Eastern Sudan [23] and other prior Ethiopian studies [1, 20, 22, 24]. The consistency could be due to the fact that contraceptive use contributes to birth spacing

**Table 6. Multivariable analysis on the determinants of short birth interval among ever married reproductive age mothers in Dessie city administration, Dessie, Ethiopia, 2019.**

| Factors | Case | Controls | Crude OR(95% CI) | p-value | AOR(95%CI) | p-value |
|---|---|---|---|---|---|---|
| Preceding pregnancy was planned | | | | | | |
| yes | 180 | 402 | 0.401(0.244–0.657) | .001 | 0.800(.348–1.839) | .599 |
| no | 38 | 34 | 1 | | 1 | |
| practice of postpartum abstinence in preceding child | | | | | | |
| yes | 161 | 359 | 0.606(0.410–0.8941) | .012 | 0.875(0.482–1.587) | .659 |
| no | 57 | 77 | 1 | | 1 | |
| ANC follow up in preceding pregnancy | | | | | | |
| Yes | 177 | 387 | 0.547(0.348–0.858) | .009 | 0.895(0.400–2.003) | 0.787 |
| No | 41 | 49 | 1 | | 1 | |
| breast fed duration from previous to last child | | | | | | |
| 0–11 | 134 | 61 | 1 | | **1** | |
| 12–23 | 64 | 197 | 0.148(0.098–0.224) | .001 | **0.291(0.154–0.550)** | **.001***  |
| > = 24 | 20 | 178 | 0.051(0.029–0.089) | .001 | **0.098(0.047–0.208)** | **.001***  |
| previous to Last child sex | | | | | | |
| male | 72 | 235 | 0.422(0.300–0.592) | 0.01 | **0.463(0.282–0.761)** | **.002***  |
| female | 146 | 201 | 1 | | 1 | |
| using any of the modern methods before the conception of your last child | | | | | 1 | |
| yes | 98 | 411 | 1 | .001 | **11.221(5.953–21.151)** | **.001***  |
| no | 120 | 25 | 20.1(12.407–32.662) | | | |
| knowledge to appropriate duration of birth interval | | | | | | |
| correctly know | 130 | 280 | 0.823(.589–1.149) | 0.253 | **0.446(0.245–0.811)** | **.008***  |
| not correctly know | 88 | 156 | 1 | | **1** | |
| Husband education | | | | | | |
| No formal education | 82 | 138 | 1.302(0.926–1.830) | 0.129 | 1.236 (0.633–2.416) | .535 |
| Had formal education | 136 | 298 | 1 | | 1 | |
| age at first marriage | | | | | | |
| less than 18 | 24 | 82 | 1 | | 1 | |
| 18–25 | 154 | 328 | 1.604(0.979–2.628) | 0.061 | 1.148(0.550–2.398) | .713 |
| Greater than 25 | 40 | 26 | 5.256(2.68–10.286) | .001 | 0.478(0.113–2.024) | .316 |
| age at first birth (years) | | | | | | |
| less than 28 | 160 | 413 | 0.154(0.092–0.257) | 0.001 | **0.363(0.166–0.793)** | **0.011***  |
| > = 28 | 58 | 23 | 1 | | **1** | |
| no of living children | | | | | | |
| 0–2 | 55 | 90 | 1 | | 1 | |
| 3–4 | 125 | 280 | 0.731(0.492–1.086) | 0.120 | .617(0.338–1.124) | .115 |
| > = 5 | 38 | 66 | 0.942(0.559–1.587) | 0.823 | 1.109(0.489–2.514) | .696 |
| Husband perception to birth spacing | | | | | | |
| Disagree strongly | 28 | 27 | 1 | | 1 | |
| Dont mind | 57 | 152 | 0.362(0.196–0.666) | 0.001 | 0.376(0.136–1.036) | .059 |
| Encouraging | 120 | 246 | 0.470(0.266–0.833) | 0.010 | 0.557(0.221–1.401) | .214 |
| Unknown | 13 | 11 | 1.140(0.436–2.980) | 0.790 | 0.873(0.195–3.908) | .859 |
| Wealth index | | | | | | |
| Poorest | 57 | 84 | 1.962(1.187–3.245) | .009. | 2.012(0.872–4.645) | .101 |
| Second | 35 | 80 | 1.265(0.733–2.183) | 0.398 | 1.486(0.606–3.647) | .387 |
| Middle | 47 | 83 | 1.638(0.976–2.747) | .062 | 2.378(1.086–5.210 | .030 |
| Fourth | 42 | 82 | 1.481(0.874–2.510) | .144 | 1.823(0.780–4.262) | 0.166 |
| Richest | 37 | 107 | 1 | | | |

*for Significant association at p<0.05)

thereby reducing the total fertility rate by different mechanisms on normal reproductive process [25].

Mothers who breastfed their preceding child for at least 24 months had lower odds of short birth interval than those who breastfed for less than 12 months. This finding was supported by different studies which revealed lengths of birth interval to be influenced by duration of breast-feeding [26–28].Moreover, studies in Arba Minch District [20] and four disadvantaged regions of Ethiopia [21] showed similar finding which may be attributed to the fact that breast feeding has contraceptive effect due to the negative hormonal feedback mechanism of the hypotha-lamic-pitutary-ovarian axis. On the contrary, according to a community based cross sectional study in Southern Ethiopia, longer duration of breast feeding was significantly associated with increased incidence of short birth interval [22].The discrepancy might be due to differences in breast-feeding practices (exclusive breastfeeding, duration and frequency of breast feeding per 24 hours) and maternal factors (age, parity, nutritional status) [12] between the two studies. Besides, methodological and other socio-cultural differences between the two study populations might have contributed for the discrepancy.

Age at first birth was an important determinant of short birth interval. Hence, the odds of short birth interval among ever married reproductive age mothers who gave their first birth at the age of ≥28 years were higher as regarded to those who gave their first birth at less than 28 years. This finding was consistent with evidences from a study in the United States [29].The consistence might be due to the reason that elderly primiparity is often considered as a possible risk factor for limited fertility and hence elderly primiparous mothers rush to complete birth-ing of all their children as narrow spaced as possible [26]. But, this study was contrary to cross-sectional studies in Bangladesh [30, 31] which revealed that mothers having first birth at higher age usually have higher birth interval. The discrepancy could be attributed to the socio cultural and methodological variations among the two study population.

The study also showed that mothers who gave male child birth had lower odds to experi-ence short birth interval than those whose child was female. This phenomenon was in line with evidences from case control studies in Arba Minch District [20] and rural pastoral com-munities of Southern Ethiopia [1]. The likely explanation of the congruence might be due to the fact that sex preference is a common culture in some communities so that giving son can be considered as a pride. Therefore, mothers who got female child from their prior birth become eager to be pregnant in short duration until they have the desired number of sons.

Mothers who knew the duration of optimum birth interval correctly had lower odds of short birth interval than those who didn't know. This finding was congruent with a case con-trol study in Arba Minch District that showed lack of information about optimal birth spacing to be an indicated reason of short birth interval [20].The likely explanation could be due to the fact that knowledge about the optimum inter birth interval is an important factor in motivat-ing mothers to utilize family planning methods and practice safe breast feeding principles thereby preventing bad obstetric outcomes of short birth interval.

Based on our findings, local health care providers (physicians, midwives, nurses and health extension workers), the city health department and policy makers should focus on different strategies for creating parental awareness about the importance of modern contraceptive use, being primiparous before 28 years old and maternal knowledge of birth spacing. Moreover, we strongly recommend that mothers should prolong their breastfeeding practice for at least two years because its effect for optimizing birth interval has been witnessed by many other studies, WHO and UNICEF [32]. However, encouraging breast feeding up to two years may not war-rant a reduction of birth interval because increasing breast feeding duration merely does not increase period of amenorrhea. This could in turn be due to differences among maternal breastfeeding practices, maternal age and parity. Women who are partially breast-feeding are

at higher risk of conceiving than women who are fully breast-feeding. The period of lactational amenorrhoea tends to be longer for older and multiparous than for younger and primiparous women. Besides, regardless of their breastfeeding practices, the other possible independent factor that may affect lactational infertility is maternal nutritional status. Therefore, despite the aforementioned confounders, maternal practice of optimal breastfeeding helps them optimize not only their health but also feto-neonatal and childhood survival.

## Strength and limitation of the study

Using community based unmatched case control study design, high response rate and inclusion of both urban and rural communities could be considered as strengths of the study.

However, mothers' failure to recall of some important determinants like their own and children's age might have introduced recall bias into the study. Besides, accessing their socially desirable answers to some questions such as history of neonatal death would have caused social desirability bias. The recall bias was dealt with enabling mothers attach their children's birth dates to unforgettable Ethiopian holidays and calendar days. Besides, it was tried to minimize social desirability bias by conducting probed maternal interviews of the events (factors) by the trained data collectors. Some factors like husbands' perception of birth spacing may not have been measured appropriately. The study lacks support of qualitative data. Moreover, the results may not be representative of the ever married women of reproductive age group in Ethiopia due to smaller sample size in this study. Besides, the association of breastfeeding duration with inter-birth interval wasn't shown by subgroups of age, parity, breast feeding practices and nutritional status of the mothers, which can be considered as a limitation of the study. All the aforementioned limitations might have attributed for less precise measurement of some factors in the study.

## Conclusion

From this study, contraceptive use, two and above years of breast feeding duration, less than 28 years of age at first birth, having male preceding child and knowing the duration of optimum birth interval correctly had significant negative odds of association with shortbirth interval.

## Supporting information

**S1 Questionnaire. Questionnaire used for data collection.**
(DOCX)

## Acknowledgments

Authors' heartfelt thank goes to data collectors, supervisors, Dessie city administration, health extension workers and study participants for their kind cooperation and invaluable collaboration.

## Author Contributions

**Conceptualization:** Habtamu Shimels Hailemeskel, Tesfaye Assebe, Tadesse Alemayehu, Demeke Mesfin Belay, Fentaw Teshome, Alemwork Baye, Wubet Alebachew Bayih.

**Data curation:** Habtamu Shimels Hailemeskel.

**Formal analysis:** Habtamu Shimels Hailemeskel, Tesfaye Assebe, Tadesse Alemayehu, Demeke Mesfin Belay, Fentaw Teshome, Alemwork Baye, Wubet Alebachew Bayih.

**Investigation:** Habtamu Shimels Hailemeskel, Tesfaye Assebe, Tadesse Alemayehu, Demeke Mesfin Belay, Fentaw Teshome, Wubet Alebachew Bayih.

**Methodology:** Habtamu Shimels Hailemeskel, Tesfaye Assebe, Tadesse Alemayehu, Demeke Mesfin Belay, Fentaw Teshome, Alemwork Baye, Wubet Alebachew Bayih.

**Software:** Habtamu Shimels Hailemeskel, Tesfaye Assebe, Demeke Mesfin Belay, Fentaw Teshome, Alemwork Baye, Wubet Alebachew Bayih.

**Supervision:** Habtamu Shimels Hailemeskel, Alemwork Baye, Wubet Alebachew Bayih.

**Validation:** Habtamu Shimels Hailemeskel, Tesfaye Assebe, Tadesse Alemayehu, Demeke Mesfin Belay, Fentaw Teshome, Wubet Alebachew Bayih.

**Visualization:** Habtamu Shimels Hailemeskel, Demeke Mesfin Belay, Fentaw Teshome.

**Writing – original draft:** Habtamu Shimels Hailemeskel, Tesfaye Assebe, Tadesse Alemayehu, Alemwork Baye.

**Writing – review & editing:** Habtamu Shimels Hailemeskel, Tesfaye Assebe, Tadesse Alemayehu, Fentaw Teshome.

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
