## [Decision Letter · Decision Letter 0]

1 May 2020

PONE-D-19-32845

Determinants of short birth interval among ever married reproductive age women: A Community based unmatched case control study at Dessie city administration, Northern Ethiopia

PLOS ONE

Dear Habtamu Shimelis,

Thank you for submitting your manuscript to PLOS ONE. After careful consideration, we feel that it has merit but does not fully meet PLOS ONE’s publication criteria as it currently stands. Therefore, we invite you to submit a revised version of the manuscript that addresses the points raised during the review process.

We would appreciate receiving your revised manuscript by 1st June, 2020. To enhance the reproducibility of your results, we recommend that if applicable you deposit your laboratory protocols in protocols.io, where a protocol can be assigned its own identifier (DOI) such that it can be cited independently in the future. For instructions see: http://journals.plos.org/plosone/s/submission-guidelines#loc-laboratory-protocols

We look forward to receiving your revised manuscript.

Kind regards,

Sharon Mary Brownie

Academic Editor

PLOS ONE

Journal Requirements:

2. Please include additional information regarding the survey or questionnaire used in the study and ensure that you have provided sufficient details that others could replicate the analyses. For instance, if you developed a questionnaire as part of this study and it is not under a copyright more restrictive than CC-BY, please include a copy, in both the original language and English, as Supporting Information. In addition, please refrain from stating p values as .000, either state the exact value or use the format p<0.001.

4. Your ethics statement must appear in the Methods section of your manuscript. If your ethics statement is written in any section besides the Methods, please move it to the Methods section and delete it from any other section. Please also ensure that your ethics statement is included in your manuscript, as the ethics section of your online submission will not be published alongside your manuscript.

Additional Editor Comments (if provided):

Reviewers have returned some substantive comments to improve your manuscript. Please carefully consider each comment and respond appropriately. Please prepare a table indicating how you have responded to each comment and please follow the advice to secure expertise in the correction of English grammar within the script.

Reviewers' comments:

Reviewer's Responses to Questions

**Comments to the Author**

1. Is the manuscript technically sound, and do the data support the conclusions?

Reviewer #1: Yes

Reviewer #2: Yes

2. Has the statistical analysis been performed appropriately and rigorously? 

Reviewer #1: Yes

Reviewer #2: Yes

3. Have the authors made all data underlying the findings in their manuscript fully available?

Reviewer #1: Yes

Reviewer #2: Yes

4. Is the manuscript presented in an intelligible fashion and written in standard English?

Reviewer #1: Yes

Reviewer #2: Yes

5. Review Comments to the Author

Reviewer #1: The authors have conducted an interesting descriptive analysis of factors that predict birth spacing of < 3 years as opposed to between 3 and 5 years in Ethiopia. They find that contraceptive use, breastfeeding, age at first birth, preceding child sex and underlying knowledge of existing advice regarding birth spacing all influenced the likelihood of a short birth interval. I have several questions for clarification.

1) You need to put the Ethiopian guidelines regarding birth spacing in a wider context in the introduction. Most high-income countries have a much shorter birth spacing. Why does Ethiopia (and a lot of other low and middle income countries) recommend a minimum of 3 years? It would benefit the reader a lot if you described the reasoning.

2) Please describe and justify the sampling frame in more detail. You should also provide the response rates for cases and controls. There should also be a figure 1 showing exactly how many cases and controls were recruited as opposed to the number included in the analysis.

3) For your power calculations, you have not described how prevalent you estimated the relevant predictors of short interpregnancy interval to be? Some of the predictors you considered are very rare and you are not adequately powered to evaluate them. You should specify the minimum prevalence of the predictors you were powered to detect in relation the estimated minimum effect size.

4) How did you decide what background factors to explore? What informed the questions that you initially decided to ask the study participants?

5) Was there really no missing information for any of the covariates? If there was any missing information in any of the covariates, how was this dealt this? I can´t see that this is described in the methods. If you have any missing data, this should be dealt with using multiple imputation.

6) Were any of the women included in the study related? I was wondering whether you have any dependencies in the data that should be dealt with in the regression analysis. For example by using robust standard errors.

7) You should clearly show the p-values from the bivariate analyses in all tables (1-4). As far as I can tell, these bivariate analyses provided the basis for your decision for to carry some covariates forward to the regression analysis.

8) You used a backwards approach to your variable selection, if I am interpreting the methods correctly. Were the results similar if you used a forward or a stepwise variable selection procedure?

9) I would recommend that you have a native English speaker help you look through the manuscript.

Reviewer #2: This was an interesting manuscript and I enjoyed reading your manuscript. However, the authors should consider addressing the following:

1. The entire manuscript needs to be revised for grammatical errors and punctuation issues. For example, on page 2, the conclusion section of the abstract, the authors started a sentence with a lower case alphabet. Also, on page 3 (introduction) the first sentence needs revising as we express maternal mortality per 100,000 live births and not "live birth".Similarly, on page 3, the last sentence in paragraph 1 needs revising particularly the sentence ...."the problem is still major public concern."

2. The authors stated on page 3 that ...."national guideline for family planning services according to the Ethiopian FMOH’s recommendation" I will recommend that the authors should provide a sentence or two from this guidelines that are applicable/relevant to their study.

3. The authors need to state the aim in the last paragraph of their introduction. At present, this is not really clear.

4. On page 8, Table 1, the authors need to correct the word 'college' which is wrongly written as 'collage'. Additionally the word 'widowed' is also wrongly written as 'windowed'

5. On page 14, Table 5, the authors need to double-check the p-values as there as selected p-values <0.05 that were not highlighted.

6. On page 15, paragraph 2, the authors should correct the word 'consistency' which was misspelt as 'consistence'.

7. The limitations of the study (page 17) needs revision and should be reported before the conclusion. Specifically, there was no mention of how the recall and social desirability bias reported in the manuscript were dealt with.

8. The authors also need to provide a few sentences on the key strength of their study.

6. PLOS authors have the option to publish the peer review history of their article (what does this mean?). If published, this will include your full peer review and any attached files.

Reviewer #1: No

Reviewer #2: No

---

## [Author Response · Author response to Decision Letter 0]

12 Oct 2020

Response letter

Dear editor

After going through the entire manuscript, you forwarded your constructive editorial comments which we missed to touch. Therefore, we are glad enough to express our sincerest thanks for your in-depth review and comments that could help improve the tone and readability of our paper.

Editor comment 1: Please ensure that your manuscript meets PLOS ONE's style requirements, including those for file naming. The PLOS ONE style templates can be found at https://journals.plos.org/plosone/s/file?id=wjVg/PLOSOne_formatting_sample_main_body.pdf andhttps://journals.plos.org/plosone/s/file?id=ba62/PLOSOne_formatting_sample_title_authors_affiliations.pdf. 

 Authors’ response: Very important comment it is! Thus, we have ensured that our manuscript meets PLOS ONE's style requirements, including those for file naming by finding the aforementioned link for PLOS ONE style templates.

Editor comment 2: Please include additional information regarding the survey or questionnaire used in the study and ensure that you have provided sufficient details that others could replicate the analyses. For instance, if you developed a questionnaire as part of this study and it is not under a copyright more restrictive than CC-BY, please include a copy, in both the original language and English, as Supporting Information. In addition, please refrain from stating p values as .000, either state the exact value or use the format p<0.001.

Authors’ response: Undoubtedly! There is a need for including the survey or questionnaire in the study. Besides, we have ensured that we have provided sufficient details that others could replicate the analyses as provided in the additional information file. We didn’t develop questionnaire as part of this study and hence no worry about copyright. In addition, we have refrained from stating p values as .000, rather we stated the exact value as it can be noticed from the yellow highlight in the revised version manuscript.

The questionnaire is newly added as additional information as it can be seen from the yellow highlight on pages 7 and 24 of the revised version manuscript.

Editor comment 3. PLOS requires an ORCID iD for the corresponding author in Editorial Manager on papers submitted after December 6th, 2016. Please ensure that you have an ORCID iD and that it is validated in Editorial Manager. To do this, go to ‘Update my Information’ (in the upper left-hand corner of the main menu), and click on the Fetch/Validate link next to the ORCID field. This will take you to the ORCID site and allow you to create a new iD or authenticate a pre-existing iD in Editorial Manager. Please see the following video for instructions on linking an ORCID iD to your Editorial Manager account: https://www.youtube.com/watch?v=_xcclfuvtxQ

 Authors’ response: the corresponding author didn’t have an ORCID iD before. Thus, based on your recommendation, the corresponding author created a new ORCID iD (0000-0002-5972-4818) in the PLOS Editorial Manager.

Editor comment 4. Your ethics statement must appear in the Methods section of your manuscript. If your ethics statement is written in any section besides the Methods, please move it to the Methods section and delete it from any other section. Please also ensure that your ethics statement is included in your manuscript, as the ethics section of your online submission will not be published alongside your manuscript.

 Authors’ response: Absolutely! It is a comment of technical relevance as per the journal requirement. Thus, our ethics statement appears in the methods section of our manuscript. Besides, our ethics statement is deleted from the declaration section and moved to the methods section as shown by the yellow highlighted text on page 8 in the ethical approval and consent to participate subsection, methods section of the revised version manuscript.

Additional Editor Comments (if provided):

Reviewers have returned some substantive comments to improve your manuscript. Please carefully consider each comment and respond appropriately. Please prepare a table indicating how you have responded to each comment and please follow the advice to secure expertise in the correction of English grammar within the script.

 Authors’ response: No doubt! We have tried our best to carefully consider each reviewer’s comment and respond appropriately. We have also followed the advice to secure expertise in the correction of English grammar within the script. Besides, we have included a point by point response letter as detailed below. 

Dear reviewer 1 

After going through the entire manuscript, you forwarded your constructive comments which we missed to touch. Therefore, we are glad enough to express our sincerest thanks for your in-depth review and comments that could help improve the tone of ourpaper.

Reviewer suggestion: The authors have conducted an interesting descriptive analysis of factors that predict birth spacing of < 3 years as opposed to between 3 and 5 years in Ethiopia. They find that contraceptive use, breastfeeding, age at first birth, preceding child sex and underlying knowledge of existing advice regarding birth spacing all influenced the likelihood of a short birth interval. I have several questions for clarification.

Authors’ response: We are really grateful for your appreciation of our efforts. Besides, we have tried our best to address all your comments point by point as detailed below. 

Reviewer comment 1: You need to put the Ethiopian guidelines regarding birth spacing in a wider context in the introduction. Most high-income countries have much shorter birth spacing. Why does Ethiopia (and a lot of other low and middle income countries) recommend a minimum of 3 years? It would benefit the reader a lot if you described the reasoning.

Authors’ response: Definitely! Description of why does Ethiopia recommend a minimum of 3 years of birth spacing in a wider context in the introduction would benefit the reader a lot for easy understanding. Therefore, the following quoted text is added to the introduction section of the revised version manuscript as it can be seen by the yellow highlighted text on pages 3 and 4, under background section. 

“Ethiopia had high population size as it was projected to reach more than 100 million and 4.0 total fertility rates in 2015. The country had also higher estimated pregnancy-related mortality ratio (PRM) of 412 deaths per 100,000 live births. Moreover, 1 in every 35 children dies within the first month; 1 in every 21 children dies before celebrating the first birthday; and 1 of every 15 children dies before reaching the fifth birthday (16). Therefore, the Ethiopian Federal Ministry of Health (FOMH) recommends spacing of childbirth at intervals of three to five years to reduce maternal, perinatal and infant mortality by optimizing the fertility rate in the country.

Reviewer comment 2) Please describe and justify the sampling frame in more detail. You should also provide the response rates for cases and controls. There should also be a figure 1 showing exactly how many cases and controls were recruited as opposed to the number included in the analysis.

Authors’ response: What a comment of relevance! It would help increase understandability of the sampling procedure! Thus, the following detail is given. 

“Multi stage sampling technique was employed to select the cases and controls. At first, 30% of the overall ‘kebeles’ (three rural and five urban kebelles), were selected by simple random sampling technique. Then, for those rural kebeles, the authors first checked family folder from health extension workers. The family folder is an extension of the Ethiopian Community Health Information System (CHIS) at the most basic level of rural health system. Health extension workers (HEWs) make individualized household family member and assessment of each household’s health behavior and assign a set of health cards for individuals in each household. We reviewed the family folder of permanently residing women in each kebele that fulfilled the inclusion criteria (less than 3 years birth interval for cases and 3–5 years’ birth interval (including 3 and 5 years for controls) by registering the birth date of the last two successive children in a family with their corresponding household identification number. However, for urban ‘kebeles’, house to house visit (census) was conducted to identify permanently residing women that fulfilled the inclusion criteria (cases and controls) by registering the birth date of the last two successive children in a family with their corresponding household identification number. Using the respective household identification number, a sampling frame of the households containing cases and controls was prepared for each kebele. Then, proportional allocation of sample size was employed to determine the study participants from each kebele. Finally, cases and controls were selected by simple random sampling technique from the existing sampling frame. Whenever more than one eligible woman was found in same selected household, only one woman was chosen by lottery method. Thus, a sample of 678 women (226 cases and 452 controls) was recruited from the sampling frame for the study. But, from these recruited 678 women, 654 women (218 cases and 436 controls) agreed to be interviewed, thereby making a response rate of 96.5% (Figure 1)”.

Amendment is located on pages 5 and 6 , methods section, subsection of sample size determination and sampling procedure in the revised version manuscript as shown by the yellow highlighted text. 

Reviewer comment:3) For your power calculations, you have not described how prevalent you estimated the relevant predictors of short inter-pregnancy interval to be? Some of the predictors you considered are very rare and you are not adequately powered to evaluate them. You should specify the minimum prevalence of the predictors you were powered to detect in relation the estimated minimum effect size.

Authors’ response: Yes indeed! For our power calculations, we have not described how prevalent we estimated the relevant predictors of short inter-pregnancy interval to be. Besides, as you said, some of the predictors we considered are rare and we are not adequately powered to evaluate them. Therefore, taking this comment and several exposure variables into account, we calculated the respective sample size just by considering the assumption of case to control ratio of 1: 2; CI: 95%; Power: 80%; minimum detectable AOR =2; design effect of 1.5 and 5% non-respondent rate. We selected the factor ‘contraceptive use’ because it yielded the maximum sample size as given in the following table (Table 1). Therefore, the sample size for this study was 678 (226 cases and 452 controls). 

Table 1: Sample size determination using different factors in the literature and the respective assumptions using Open EPI INFO version 7 software.

Factors Assumption Total sample size References 

Contraceptive user P of exposure in controls =66.7%

 678 (Hailu and Gulte, 2016)

Residence/urban P of exposure in controls =52.1%

 540 (Yohannes et al., 2011)

Husbands’ occupation /Employee P of exposure in controls =51.7%

 537 (Yohannes et al., 2011)

Mothers’ education /Has formal education P of exposure in controls =48.3%

 524 (Hailu and Gulte, 2016)

Parity />=5 children P of exposure in controls = 49.2% 524 (Begna Z. et al., 2013)

Sex of the index child /male P of exposure in controls = 64.2%

 638 (Begna Z. et al., 2013)

Age of the mother/ 25-29 P of exposure in controls = 24.9%

 576 (Begna Z. et al., 2013)

Status of index child /Alive P of exposure in controls = 41.3%

 509 (Tsegaye Dereje et al., 2017)

Wealth index/ Richest P of exposure in controls = 25.2%

 509 (Hailu and Gulte, 2016)

Amendment is located on pages 5 and 6 in the revised version manuscript as shown by the yellow highlighted text. 

Reviewer comment 4) How did you decide what background factors to explore? What informed the questions that you initially decided to ask the study participants?

Authors’ response: After reviewing different literature (Japheth Osotsi Awiti, 2013; Hailu and Gulte, 2016; Ayanaw A., 2008; Baschieri and Hinde, 2007, Begna Z. et al., 2013; Central Statistical Agency (CSA) [Ethiopia] and ICF, 2016; Hailu and Gulte, 2016; Tsegaye Dereje et al., 2017; Yohannes et al., 2011) that addressed proximate, intermediate, socio- demographic and economic determinants of birth interval, we decided the background factors to be explored in this study. 

Reviewer comment 5) Was there really no missing information for any of the covariates? If there was any missing information in any of the covariates, how was this dealt this? I can´t see that this is described in the methods. If you have any missing data, this should be dealt with using multiple imputations.

Authors’ response: There is no missing information for any of the covariates in this study. This was because incomplete questionnaires were returned to the data collectors for completion by referring to the respective household identification number on a daily basis of checking all the questionnaires.

Amendment is located on page 7, methods section, subsection of data quality control, in the revised version manuscript as shown by the yellow highlighted text. 

Reviewer comment 6) Were any of the women included in the study related? I was wondering whether you have any dependencies in the data that should be dealt with in the regression analysis. For example by using robust standard errors.

Authors’ response: What a comment of paramount importance! Answering this comment helps assure data quality and management of bias. As mentioned in the sampling procedure, it was tried to minimize bias from intra-cluster correlation effect (dependencies) by selecting only one of the eligible women in a selected household. Besides, standard error was used during multivariate regressions and there was no any factor whose standard error greater than two indicating no dependency between mothers regarding the considered factors. 

Amendment is located on page 8, methods section, subsection of data processing and analysis, in the revised version manuscript as shown by the yellow highlighted text. 

Reviewer comment 7) You should clearly show the p-values from the bivariate analyses in all tables (1-4). As far as I can tell, these bivariate analyses provided the basis for your decision for to carry some covariates forward to the regression analysis.

Authors’ response: Undoubtedly! Considering your constructive comment, we have now clearly shown the p-values from the bivariate analyses in all tables (former tables 1-4, currently tables 2-5). As you said, these bivariate analyses provided the basis for our decision to carry some covariates forward to the regression analysis. The detailed response is given below and also shown by the yellow highlighted column of the P-value for tables 2-5.

Table 2: Socio-demographic characteristics on short birth interval among ever married mothers (case=218, control=436) in Dessie city administration, Dessie , Ethiopia 2019.

Amendment is located on page , line of the revised version manuscript as shown by the yellow highlighted text. 

Factors Category Case (%) Control(%) P value

Rsidence Urban 130(59.7%) 273(62.65) 0.460

 Rural 88(40.3%) 163(37.4%) 

Marital status Married 186(85.3%) 364(83.5%) 0.759

 Divorced 21(9.6%) 44(10.1%) 

 Widowed 11(5.1%) 28(6.4%) 

Religion Orthodox 92(42.2%) 173(39.7%) 0.287

 Muslim 124(56.9%) 249(57.1%) 

 Protestant 2(0.9%) 14(3.2%) 

Ethinicity Amhara 200(91.7%) 399(91.5%) 0.926

 Tgrai 7(3.2%) 11(2.5%) 

 Oromo 6(2.7%) 14(3.2%) 

 Others1 5(2.3%) 12(2.8%) 

Mother’s education No formal education 45(20.6%) 70(16.1%) 0.546

 read and write 42(19.3%) 86(19.7%) 

 Elementary 34(15.6%) 81(18.6%) 

 Secondary 39(17.9%) 89(20.4%) 

 Collage and above 58(26.6%) 110(25.2%) 

Husband education No formal education 50(22.9%) 69(15.8%) 0.104

 read and write 32(14.7%) 69(15.8%) 

 Elementary 13(5.9%) 42(9.6%) 

 Secondary 41(18.8%) 72(16.5%) 

 College and above 82(37.6%) 184(42.2%) 

Mothers’ occupation employee(GO/NGO) 43(19.7%) 91(20.9%) 0.730

 house wife 125(57.3%) 232(53.2%) 

 Merchant 28(12.8%) 53(12.2%) 

 Student 9(4.1%) 29(6.7%) 

 Farmer 10(4.6%) 19(4.4%) 

 daily workers 3(1.4%) 11(2.5%) 

 Others2 0(0%) 1(0.2%) 

Husband occupation employee(GO/NGO) 84(38.5%) 164(37.6%) 0.086

 Merchant 66(30.3%) 129(29.6%) 

 Student 0(0%) 2(0.5%) 

 Farmer 63(28.9%) 107(24.5%) 

 daily workers 4(1.8%) 23(5.3%) 

 Others3 1(0.5%) 11(2.5%) 

 Number of wives

wealth index One 216(99.1%) 434(99.5%) 0.478

 More than one 2(0.9%) 2(0.5%) 

 Poorest 57(26.1%) 84(19.3%) 0.096

 Second 35(16.1%) 80(18.3%) 

 Middle 47(26.6%) 83(19.0%) 

 Fourth 42(19.3%) 82(18.8%) 

 Richest 37(17.0%) 107(24.5%) 

1Afar, Gurage2 House servant,3Religious leader

Table 3: Knowledge and attitude of birth interval among ever married reproductive age mothers (case=218, control=436) in Dessie city administration, Dessie, Ethiopia 2019.

Factors Category Case (%) Control (%) P value

Heard about optimal birth interval Yes 165(75.7%) 352(80.7%) 0.336

 No 53(24.3%) 84(19.3%) 

Optimum number of years between two successive births Below three years 19(11.5) 46(13.1%) 0.701

 Three to five years 130(78.8%) 280(79.5%) 

 Above five years 13(7.8%) 23(6.5%) 

 I am not sure 3(1.8%) 3(0.8%) 0.562

A minimum of 3 years of birth interval is essential between two successive births Strongly agree 81(37.2%) 139(31.9%) 

 Agree 134(61.5%) 291(66.7%) 

 no idea 2(0.9%) 3(0.7%) 

 Disagree 1(0.5%) 3(0.7%) 

Husband's perception regarding birth spacing Disagree strongly 28(12.8%) 27(6.2%) 0.001

 don't mind 57(26.1%) 152(34.9%) 

 Encouraging 120(55.04%) 246(56.4%) 

 Unknown 13(5.96%) 11(2.5%) 

External influences to give birth in short interval My family 37(16.97%) 61(13.99%) 0.258

 Mother in law 21(9.63%) 60(13.76%) 

 Father in law 7(3.2%) 12(2.75%) 

 Societies norm 9(4.1%) 5(1.1%) 

 None 144(66.1%) 298(68.4%) 

Perceived advantages of optimum birth spacing Yes 205(94.04%) 406(93.1%) 0.655

 No 13(5.96%) 30(6.9%) 

Perceived disadvantages of short birth interval Yes 204(93.6%) 404(92.7%) 0.665

 No 14(6.4%) 32(7.3%) 

Table 4: Obstetrics related factors of short birth interval among ever married reproductive age mothers (case=218, control=436)inDessie city administration, Dessie, Ethiopia 2019.

Factors Category Case (%) Control (%) P value

Fetal outcome of first delivery Live birth 191(87.6%) 382(87.62%) 0.352

 still birth 11(5.04%) 13(2.98%) 

 Abortion 3(1.4%) 13(2.98%) 

 Neonatal mortality 13(5.96%) 28(6.42%) 

Prior history of infertility Yes 4(1.83%) 3(0.69%) 0.279

 No 214(98.17%) 433(99.31%) 

Ever given birth to any child who died Yes 31(14.2%) 58(13.3%) 0.723

 No 187(85.8%) 378(86.7%) 

Male to female ratio of living children More than one 71(32.6%) 160(36.7%) 0.355

 One 63(28.89%) 135(30.96%) 

 Less than one 49(22.48%) 74(16.97%) 

 Males only 15(6.9%) 36(8.26%) 

 Females only 20(9.17%) 31(7.11%) 

Previous to last pregnancy is planned Yes 180(82.6%) 402(92.2%) 0.001

 No 38(17.4%) 34(7.8%) 

Practice postpartum abstinence before the last child Yes 161(73.85%) 359(82.3%) 0.011

 No 57(26.15%) 77(17.7%) 

Mode of delivery of previous to last birth Vaginal delivery 197(90.4%) 392(89.9%) 0.981

 Cesarean section 14(6.4%) 29(6.7%) 

 Instrumental delivery 7(3.2%) 15(3.4%) 

ANC follow up in preceding pregnancy Yes 172(78.9%) 387(88.8%) 0.009

 No 46(21.1%) 49(11.2%) 

Place of delivery of previous to last birth Home 25(11.5%) 39(8.9%) 0.308

 Health institution 193(88.5%) 397(91.1%) 

Pattern of menstruation in previous to last deliveries Regular 185(84.9%) 362(83.02%) 0.550

 Irregular 33(15.1%) 74(16.97%) 

Ever had chronic diseases (HTN,DM ,others) before the last child Yes 16(7.3%) 44(10.1%) 0.255

 No 202(92.7%) 392(89.9%) 

Ever had history of postpartum complications in previous to last deliveries Yes 26(11.9%) 61(13.99%) 0.464

 No 192(88.1%) 375(86.01%) 

Last child sex Male 121(55.5%) 238(54.6%) 0.824

 Female 97(44.5%) 198(45.4%) 

Is last child alive Yes 217(99.5%) 434(99.5%) 0.741

 No 1(0.5%) 2(0.5%) 

previous to last child sex Male 72(33%) 235(53.9%) 0.001

 Female 116(53.2%) 201(46.1%) 

Is previous to last child alive Yes 215(98.6%) 434(99.5%) 0.254

 No 3(1.4%) 2(0.5%) 

Parity <5 180 (82.5%) 370(84.8%) 0.450

 >=5 38(17.5%) 66(15.2%) 

Table 5: Breast feeding duration and contraceptive use among ever married reproductive age mothers in Dessie city administration, Dessie, Ethiopia 2019 

Factors Category Case (%) Control(%) P value

Did you breast feed previous to last child Yes 152(69.7%) 400(91.7%) 0.001

 No 66(30.3%) 36(8.3%) 

Did you exclusively breastfeed previous to last child Yes 80(52.6%) 295(73.8%) 0.001

 No 72(47.4%) 105(26.2) 

 Breast feeding duration 0-11 134(61.5%) 61(13.99%) 0.001

 12-23 64(29.4%) 197(45.18%) 

 >=24 20(9.2%) 178(40.83%) 

Using any of the modern methods before the conception of your last child Yes 98(44.95%) 411(94.3%) 0.001

 No 120(55.05%) 25(5.7%) 

Decision maker about Family planning Self 104(47.7%) 190(43.58) 

 Both husband and wife 97(44.5%) 227(52.06%) 0.261

 Husband only 3(1.4%) 13(2.98%) 

 No one 14(6.4%) 6(1.38%) 

Perception of family planning method Agree 183(83.9%) 428(98.2%) 0.001

 Disagree 34(15.6%) 4(0.9%) 

 Neutral 1(0.5%) 4(0.9%) 

Distance from health institution Less than 30 minutes 93(42.7%) 197(45.2%) 0.799

 30-1hrs 123(56.4%) 236(54.1%) 

 Greater than 1 hr 2(0.9%) 3(0.7%) 

Reviewer comment 8) You used a backwards approach to your variable selection, if I am interpreting the methods correctly. Were the results similar if you used a forward or a stepwise variable selection procedure?

Authors’ response: We used backward stepwise LR to identify variables which had the largest contribution to the model. The result in forward or a stepwise variable selection method was similar on significance of the variables, but little change in adjusted odds ratio, p value and confidence interval were observed. 

Amendment is located on page 14, results section, subsection of determinants of short birth interval as shown by the yellow highlighted text in the revised version manuscript. 

Reviewer comment 9) I would recommend that you have a native English speaker help you look through the manuscript.

Authors’ response: what a similar comment with reviewer two. Therefore, similar response is given as mentioned below in the quoted text.

“From repeated proof-reading of the manuscript, we found several grammatical errors, interlinings, police titles, punctuation errors, wordings and spelling errors. Therefore, finding our colleague who has Master of Arts in English, we have tried our best to thoroughly copyedit the manuscript for English language usage. These changes are found throughout the revised version manuscript.”

Dear reviewer #2

After going through the entire manuscript, you forwarded your constructive comments which we missed to touch. Therefore, we are glad enough to express our sincerest thanks for your in-depth review and comments that could help improve the tone of our paper.

Reviewer suggestion: This was an interesting manuscript and I enjoyed reading your manuscript. However, the authors should consider addressing the following

Authors’ response: We are really grateful for your appreciation of our efforts. Besides, we have tried our best to address all your comments point by point as detailed below. 

Reviewer comment 1: The entire manuscript needs to be revised for grammatical errors and punctuation issues. For example, on page 2, the conclusion section of the abstract, the authors started a sentence with a lower case alphabet. Also, on page 3 (introduction) the first sentences need revisiting as we express maternal mortality per 100,000 live births and not "live birth". Similarly, on page 3, the last sentence in paragraph 1 needs revising particularly the sentence ...."the problem is still major public concern."

Authors’ response: Sure! From repeated proof-reading of the manuscript, we found several grammatical errors, interlinings, police titles, punctuation errors, wordings and spelling errors. Therefore, finding our colleague who has Master of Arts in English, we have tried our best to thoroughly copyedit the manuscript for English language usage. These changes are found throughout the revised version manuscript. Moreover, the aforementioned reviewer’s specific concerns are addressed as listed below.

Conclusion: Contraceptive use, breast feeding duration, age at first birth, preceding child sex and knowing the duration of birth interval correctly were independent determinants of short birthinterval.

Inter birth interval refers to the time interval from one child’s birth date until the next child’s birth date between two consecutive live births.

Amendment is located on pages 2 and 3 of the revised version manuscript as shown by the yellow highlighted text. 

Reviewer comment 2. The authors stated on page 3 that ...."national guideline for family planning services according to the Ethiopian FMOH’s recommendation" I will recommend that the authors should provide a sentence or two from this guidelines that are applicable/relevant to their study.

Authors’ response: Considering the given comment, the following sentence was taken from the Ethiopian national guideline of family planning services.

 “The Ethiopian Federal Ministry of Health (FOMH) recommends spacing of childbirth at intervals of three to five years to reduce maternal, perinatal and infant mortality by optimizing the fertility rate in the country.”

Amendment is located on page 3, paragraph 3 of the revised version manuscript as shown by the yellow highlighted text. 

Reviewer comment 3. The authors need to state the aim in the last paragraph of their introduction. At present, this is not really clear.

Authors’ response: Absolutely! At present, the aim of this study is not really clear. Therefore, it is now stated in the last paragraph of the introduction concisely as shown by the yellow highlighted text on page 4 of the revised version manuscript. The aim is also given below. 

“Therefore, this study was aimed at identifying factors that have significant odds of association with short inter-birth interval among a community-based sample of Ethiopian women in Dessie city administration, 2019.”

Reviewer comment 4. On page 8, Table 1, the authors need to correct the word 'college' which is wrongly written as 'collage'. Additionally the word 'widowed' is also wrongly written as 'windowed'

Authors’ response: Certainly! The misspelt words are corrected accordingly as shown by the yellow highlighted text in table 2 of the revised version manuscript which is also given below.

Table 2: Socio-demographic characteristics on short birth interval among ever married mothers (case=218, control=436) in Dessie city administration, Dessie , Ethiopia 2019

Factors Category Case (%) Control(%)

Rsidence Urban 130(59.7%) 273(62.65)

 Rural 88(40.3%) 163(37.4%)

Marital status Married 186(85.3%) 364(83.5%)

 Divorced 21(9.6%) 44(10.1%)

 Widowed 11(5.1%) 28(6.4%)

Religion Orthodox 92(42.2%) 173(39.7%)

 Muslim 124(56.9%) 249(57.1%)

 Protestant 2(0.9%) 14(3.2%)

Ethinicity Amhara 200(91.7%) 399(91.5%)

 Tgrai 7(3.2%) 11(2.5%)

 Oromo 6(2.7%) 14(3.2%)

 Others1 5(2.3%) 12(2.8%)

Mother’s education No formal education 45(20.6%) 70(16.1%)

 read and write 42(19.3%) 86(19.7%)

 Elementary 34(15.6%) 81(18.6%)

 Secondary 39(17.9%) 89(20.4%)

 Collage and above 58(26.6%) 110(25.2%)

Husband education No formal education 50(22.9%) 69(15.8%)

 read and write 32(14.7%) 69(15.8%)

 Elementary 13(5.9%) 42(9.6%)

 Secondary 41(18.8%) 72(16.5%)

 College and above 82(37.6%) 184(42.2%)

Mothers’ occupation employee(GO/NGO) 43(19.7%) 91(20.9%)

 house wife 125(57.3%) 232(53.2%)

 Merchant 28(12.8%) 53(12.2%)

 Student 9(4.1%) 29(6.7%)

 Farmer 10(4.6%) 19(4.4%)

 daily workers 3(1.4%) 11(2.5%)

 Others2 0(0%) 1(0.2%)

Husband occupation employee(GO/NGO) 84(38.5%) 164(37.6%)

 Merchant 66(30.3%) 129(29.6%)

 Student 0(0%) 2(0.5%)

 Farmer 63(28.9%) 107(24.5%)

 daily workers 4(1.8%) 23(5.3%)

 Others3 1(0.5%) 11(2.5%)

 Number of wives

wealth index One 216(99.1%) 434(99.5%)

 More than one 2(0.9%) 2(0.5%)

 Poorest 57(26.1%) 84(19.3%)

 Second 35(16.1%) 80(18.3%)

 Middle 47(26.6%) 83(19.0%)

 Fourth 42(19.3%) 82(18.8%)

 Richest 37(17.0%) 107(24.5%)

1Afar, Gurage2 House servant,3Religious leader

Amendment is located on page 9, in table 2 , results section, sociodemographic characteristics subsection of the revised version manuscript as shown by the yellow highlighted text. 

Reviewer comment 5. On page 14, Table 5, the authors need to double-check the p-values as there as selected p-values <0.05 that were not highlighted.

Authors’ response: Based on the given comment, Table 5 (currently, table 6) has been double-checked if there is any P-value <0.05 that were not highlighted. Thus, the double-checked table is given below with yellow highlight of P-values that were not highlighted.

Table 6: Multivariable analysis on the determinants of short birth interval among ever married reproductive age mothers in Dessie city administration, Dessie, Ethiopia 2019. 

Factors Case Controls Crude OR(95% CI) p-value AOR(95%CI) p-value

Preceding pregnancy was planned

yes

no 

180

38 

402

34 

0.401(0.244-0.657)

1 

.001 

0.800 (.348-1.839)

1 

.599

practice of postpartum abstinence in preceding child

yes

no 

161

57 

359

77 

0.606(0.410-0.8941)

1 

.012

0.875(0.482-1.587)

1 

.659

ANC follow up in preceding pregnancy

Yes

No 

177

41 

387

49 

0.547(0.348-0.858)

1 

.009 

0.895(0.400-2.003)

1 

0.787

breast fed duration from previous to last child

0-11

12-23

>=24 

134

64

20 

61

197

178 

1

0.148(0.098-0.224)

0.051(0.029-0.089) 

.001

.001 

1

0.291(0.154-0.550)

0.098(0.047-0.208) 

.001*

.001*

previous to Last child sex

male

female 

72

146 

235

201 

0.422(0.300-0.592)

1 

0.01 

0.463(0.282-0.761)

1 

.002*

using any of the modern methods before the conception of your last child

yes

no 

98

120 

411

25 

1

20.1(12.407-32.662) 

.001 

1

11.221(5.953-21.151) 

.001*

knowledge to appropriate duration of birth interval 

correctly know 

not correctly know 

130

88 

280

156 

0.823(.589-1.149)

1 

0.253

0.446(0.245-0.811)

1 

.008*

Husband education

No formal education

Had formal education 

82

136 

138

298 

1.302(0.926-1.830)

1 

0.129 

1.236 (0.633-2.416)

1 

.535

age at first marriage

less than 18

18-25

Greater than 25 

24

154

40 

82

328

26 

1

1.604(0.979-2.628)

5.256(2.68-10.286) 

0.061

.001 

1 

1.148(0.550-2.398)

0.478(0.113-2.024) 

.713

.316

age at first birth (years) 

less than 28

>=28 

160

58 

413

23 

0.154(0.092-0.257)

1 

0.001

0.363(0.166-0.793)

1 

0.011*

no of living children

 0-2

3-4

>=5 

55

125

38 

90

280

66 

1

0.731(0.492-1.086)

0.942(0.559-1.587) 

0.120

0.823

1 

.617(0.338-1.124)

1.109(0.489-2.514) 

.115

.696

Husband perception to birth spacing

Disagree strongly

Dont mind

Encouraging

Unknown 

28

57

120

13 

27

152

246

11 

1

0.362(0.196-0.666)

0.470(0.266-0.833)

1.140(0.436-2.980) 

0.001

0.010

0.790 

1

0.376(0.136-1.036)

0.557(0.221-1.401)

0.873(0.195-3.908) 

.059

.214

.859

Wealth index

Poorest 

Second

Middle

Fourth 

57

35

47

42 

84

80

83

82 

1.962(1.187-3.245)

1.265(0.733-2.183)

1.638(0.976-2.747)

1.481(0.874-2.510) 

.009.

0.398

.062

.144 

2.012(0.872-4.645)

1.486(0.606-3.647)

2.378(1.086-5.210

1.823(0.780-4.262)

.101

.387

.030

0.166

Richest 37 107 1

*for Significant association at p<0.05)

Amendment is located in table 6, on pages 15 and 16 of the revised version manuscript as shown by the yellow highlighted text. 

Reviewer comment 6: On page 15, paragraph 2, the authors should correct the word 'consistency' which was misspelt as 'consistence'.

Authors’ response: Yes indeed! On page 15, paragraph 2 of the original document, the word 'consistency' was misspelt as 'consistence'. Thus, it has been corrected as listed below.

“The consistency could be due to the fact that contraceptive use contributes to birth spacing thereby reducing the total fertility rate by different mechanisms on normal reproductive process [25].”

Amendment is located on page 17, first paragraph of the revised version manuscript as shown by the yellow highlighted text. 

Reviewer comment 7. The limitations of the study (page 17) needs revision and should be reported before the conclusion. Specifically, there was no mention of how the recall and social desirability bias reported in the manuscript were dealt with.

Authors’ response: Quite sure! The limitations of the study (page 17 of the original document), is currently revised and reported before the conclusion. Specifically, a mention of how the recall and social desirability bias reported in the manuscript were dealt with is stated clearly as given below. 

 “The recall bias was dealt with enabling mothers attach their children’s birth dates to unforgettable Ethiopian holidays and calendar days. Besides, it was tried to minimize social desirability bias by conducting probed maternal interviews of the events (factors) by the trained data collectors.”

Amendment is located on 18, strength and limitation section of the revised version manuscript as shown by the yellow highlighted text. 

Reviewer comment 8. The authors also need to provide a few sentences on the key strength of their study.

Authors’ response: Certainly! The key strength of our study was “Using community based unmatched case control study design, high response rate and inclusion of both urban and rural communities.”

Amendment is located on page 18, strength and limitation section of the revised version manuscript as shown by the yellow highlighted text. 

We look forward to receiving your immediate and kind response!

Habtamu Shimelis Hailemeskel (On behalf of all authors)

---

## [Decision Letter · Decision Letter 1]

9 Nov 2020

PONE-D-19-32845R1

Determinants of short birth interval among ever married reproductive age women: A Community based unmatched case control study at Dessie city administration, Northern Ethiopia

PLOS ONE

Dear Dr. Habtamu Shimelis,

Thank you for submitting your manuscript to PLOS ONE. After careful consideration, we feel that it has merit but does not fully meet PLOS ONE’s publication criteria as it currently stands. Therefore, we invite you to submit a revised version of the manuscript that addresses the points raised during the review process.

We look forward to receiving your revised manuscript.

Kind regards,

Sharon Mary Brownie

Academic Editor

PLOS ONE

Reviewers' comments:

Reviewer's Responses to Questions

**Comments to the Author**

1. If the authors have adequately addressed your comments raised in a previous round of review and you feel that this manuscript is now acceptable for publication, you may indicate that here to bypass the “Comments to the Author” section, enter your conflict of interest statement in the “Confidential to Editor” section, and submit your "Accept" recommendation.

Reviewer #1: (No Response)

Reviewer #2: All comments have been addressed

2. Is the manuscript technically sound, and do the data support the conclusions?

Reviewer #1: Partly

Reviewer #2: Yes

3. Has the statistical analysis been performed appropriately and rigorously? 

Reviewer #1: Yes

Reviewer #2: Yes

4. Have the authors made all data underlying the findings in their manuscript fully available?

Reviewer #1: Yes

Reviewer #2: Yes

5. Is the manuscript presented in an intelligible fashion and written in standard English?

Reviewer #1: Yes

Reviewer #2: Yes

6. Review Comments to the Author

Reviewer #1: The authors have adressed most of my comments. I only have a few minor points.

1) Delete this text from the conclusion:

"We want to emphasize that our study involved neither experimental nor observational design and

hence our recommendations are not based on causal mechanisms. Our recommendations are

rather based on the assumption that short birth interval is a potentially modifiable risk factor of

adverse pregnancy outcomes. Hence, intervening on its identified independent predictors

(significant factors) helps optimize inter-birth interval."

Your study is observational and this suggested deleted text does not add any useful information.

2) Move the last paragraph away from the conclusion section to the discisttion:

"Based on our findings, local health care providers (physicians, midwives, nurses and health

extension workers), the city health department and policy makers should focus on different

strategies for creating parental awareness about the importance of modern contraceptive use,

being primiparous before 28 years old and maternal knowledge of birth spacing. Moreover, we

strongly recommend that mothers should prolong their breastfeeding practice for at least two

years because its effect for optimizing birth interval has been witnessed by many other studies,

WHO and UNICEF. However, encouraging breast feeding up to two years may not warrant a

reduction of birth interval because increasing breast feeding duration merely does not increase

period of amenorrhea. This could in turn be due to differences among maternal breastfeeding

practices, maternal age and parity. Women who are partially breast-feeding are at higher risk of

conceiving than women who are fully breast-feeding. The period of lactational amenorrhoea

tends to be longer for older and multiparous than for younger and primiparous women. Besides,

regardless of their breastfeeding practices, the other possible independent factor that may affect

lactational infertility is maternal nutritional status. Therefore, despite the aforementioned

confounders, maternal practice of optimal breastfeeding helps them optimize not only their

health but also feto-neonatal and childhood survival."

3) You can consider shortening the section on the sample size determination and selection procedure, and the section on data quality control.

Reviewer #2: The authors have meticulously addressed my comments and the manuscript has been further strengthened.

7. PLOS authors have the option to publish the peer review history of their article (what does this mean?). If published, this will include your full peer review and any attached files.

Reviewer #1: No

Reviewer #2: No

---

## [Author Response · Author response to Decision Letter 1]

13 Nov 2020

Dear academic Editor (Sharon Mary Brownie)

After going through the entire revised version manuscript, you forwarded your constructive editorial comment. Therefore, we are glad enough to express our sincerest thanks for your helpful comment that could help improve the tone and readability of our paper.

Editor comment: Thank you for submitting your manuscript to PLOS ONE. After careful consideration, we feel that it has merit but does not fully meet PLOS ONE’s publication criteria as it currently stands. Therefore, we invite you to submit a revised version of the manuscript that addresses the points raised during the review process.

Authors’ response: We are really delighted with your constructive editorial comments. Hence, we have addressed the points raised during the review process and the corrections are incorporated within the second round revised version manuscript. All the improved changes are shown by tracked insertions and deletions.

Dear reviewer 1 

After going through the entire manuscript, you forwarded your constructive comments which we missed to touch. Therefore, we are glad enough to express our sincerest thanks for your in-depth review and comments that could help improve the tone of our paper.

Reviewer comment: The authors have addressed most of my comments. I only have a few minor points.

Authors’ response: We are most grateful for your acknowledgment of our efforts in addressing most of your comments. Besides, we have tried to address your current concerns point by point as detailed below. 

Reviewer comment: 1) Delete this text from the conclusion:

"We want to emphasize that our study involved neither experimental nor observational design and hence our recommendations are not based on causal mechanisms. Our recommendations are

rather based on the assumption that short birth interval is a potentially modifiable risk factor of

adverse pregnancy outcomes. Hence, intervening on its identified independent predictors 

(significant factors) helps optimize inter-birth interval."

Your study is observational and this suggested deleted text does not add any useful information.

Authors’ response: We strongly agree with the reviewer’s comment and hence the aforementioned text has been deleted from the conclusion section as it can be appreciated from the tracked deletion, paragraph 1 of the conclusion, on page 19 of the revised version manuscript.

Reviewer comment: 2) Move the last paragraph away from the conclusion section to the discussion:

"Based on our findings, local health care providers (physicians, midwives, nurses and health

extension workers), the city health department and policy makers should focus on different

strategies for creating parental awareness about the importance of modern contraceptive use,

being primiparous before 28 years old and maternal knowledge of birth spacing. Moreover, we

strongly recommend that mothers should prolong their breastfeeding practice for at least two

years because its effect for optimizing birth interval has been witnessed by many other studies,

WHO and UNICEF. However, encouraging breast feeding up to two years may not warrant a

reduction of birth interval because increasing breast feeding duration merely does not increase

period of amenorrhea. This could in turn be due to differences among maternal breastfeeding

practices, maternal age and parity. Women who are partially breast-feeding are at higher risk of

conceiving than women who are fully breast-feeding. The period of lactational amenorrhoea

tends to be longer for older and multiparous than for younger and primiparous women. Besides,

regardless of their breastfeeding practices, the other possible independent factor that may affect

lactational infertility is maternal nutritional status. Therefore, despite the aforementioned

confounders, maternal practice of optimal breastfeeding helps them optimize not only their

health but also feto-neonatal and childhood survival."

Authors’ response: Well! We are grateful for your comment of importance. Thus, the last paragraph has been moved away from the conclusion section to the discussion as shown by the tracked insertions and deletions in the discussion and conclusion sections of the revised version manuscript on pages 18 and 19. 

Reviewer comment: 3) You can consider shortening the section on the sample size determination and selection procedure, and the section on data quality control.

Authors’ response: Great thanks! Based on the given comment, the section on the sample size determination and selection procedure, and the section on data quality control have been shortened to a reasonable extent. The amendment can be appreciated from the tracked deletion on pages 5, 6 and 7, in the methods section, subsection of sample size determination and selection procedure and data quality control.

---

## [Editor Report · Decision Letter 2]

16 Nov 2020

Determinants of short birth interval among ever married reproductive age women: A Community based unmatched case control study at Dessie city administration, Northern Ethiopia

PONE-D-19-32845R2

Dear Dr. Habtamu Shimelis,

We’re pleased to inform you that your manuscript has been judged scientifically suitable for publication and will be formally accepted for publication once it meets all outstanding technical requirements.

Kind regards,

Sharon Mary Brownie

Academic Editor

PLOS ONE
---

## [Editor Report · Acceptance letter]

20 Nov 2020

PONE-D-19-32845R2 

Determinants of short birth interval among ever married reproductive age women: A Community based unmatched case control study at Dessie city administration, Northern Ethiopia 

Dear Dr. Shimels Hailemeskel:

I'm pleased to inform you that your manuscript has been deemed suitable for publication in PLOS ONE. Congratulations! Your manuscript is now with our production department. 

Kind regards, 

on behalf of

Professor Sharon Mary Brownie 

Academic Editor

PLOS ONE